# Creating a general-purpose generative model for healthcare data based on multiple clinical studies

Hiroshi Maruyama[1,2,3☯*], Kotatsu Bito[4☯], Yuki Saito[4], Masanobu Hibi[5], Shun Katada[5], Aya Kawakami[4], Kenta Oono[2], Nontawat Charoenphakdee[2], Zhengyan Gao[2], Hideyoshi Igata[2], Masashi Yoshikawa[2], Yoshiaki Ota[2], Hiroki Okui[2], Kei Akita[2], Shoichiro Yamaguchi[2], Yohei Sugawara[2], Shin-ichi Maeda[2]

1 Kao Corporation, Chuo-ku, Tokyo, Japan, 2 Preferred Networks Inc., Chiyoda-ku, Tokyo, Japan, 3 Research into Artifacts, Center for Engineering, The University of Tokyo, Bunkyo-ku, Tokyo, Japan, 4 Digital Business Creation, Kao Corporation, Chuo-ku, Tokyo, Japan, 5 Human Health Care Product Research Laboratories, Kao Corporation, Sumida-ku, Tokyo, Japan

☯ These authors contributed equally to this work.
* maruyama@acm.org

**Data availability statement:** For Data Sources A and C, the Institutional Review Board (IRB) of Kao Corporation determined that unrestricted public sharing of the datasets is not permissible

## Abstract

Data for healthcare applications are typically customized for specific purposes but are often difficult to access due to high costs and privacy concerns. Rather than prepare separate datasets for individual applications, we propose a novel approach: building a general-purpose generative model applicable to virtually any type of healthcare application. This generative model encompasses a broad range of human attributes, including age, sex, anthropometric measurements, blood components, physical performance metrics, and numerous healthcare-related questionnaire responses. To achieve this goal, we integrated the results of multiple clinical studies into a unified training dataset and developed a generative model to replicate its characteristics. The model can estimate missing attribute values from known attribute values and generate synthetic datasets for various applications. Our analysis confirmed that the model captures key statistical properties of the training dataset, including univariate distributions and bivariate relationships. We demonstrate the model's practical utility through multiple real-world applications, illustrating its potential impact on predictive, preventive, and personalized medicine.

## Author summary

Digital technologies are expected to revolutionize healthcare, yet digital healthcare has not reached its full potential. A major bottleneck is the poor data availability. Due to concerns regarding privacy and cost, healthcare data is very difficult to access. Here, our aim was to provide a general-purpose statistical model that can be used in place of actual data. Recent advancements in machine-learning technology, especially in *generative*

owing to ethical considerations. Moreover, the participants of these studies did not provide consent for public release, even after anonymization. Data Source B consists of third-party datasets that were commercially licensed from a private data aggregator in Japan under contracts that explicitly prohibit public redistribution. Owing to these ethical and contractual restrictions, the underlying individual-level data cannot be made publicly available. Researchers who wish to request access to the data should contact the Digital Business Creation division at Kao Corporation (kaodbc-contact@kao.com). Requests will be considered in consultation with the relevant ethical committees and IRBs, and access may be granted subject to appropriate agreements and approvals. The statistical properties of the data are approximated in the model and are available via an application program interface (API).

**Funding:** This study was solely sponsored by Kao Corporation (Tokyo, Japan), which provided full financial support for the entire research project. Kao Corporation covered all costs associated with project management, data collection, and computational resources for model development of Preferred Networks, Inc. (Tokyo, Japan), which was commissioned as a model development partner. The funder had no role in study design, data collection and analysis, decision to publish, or preparation of the manuscript. (Funder website: https://www.kao.com/). None of the authors have received any other specific funding for this study.

**Competing interests:** I have read the journal's policy and the authors of this manuscript have the following competing interests: KB, Y Saito, MH, SK, and AK are employees of Kao Corporation (Tokyo, Japan). KO, NC, ZG, HI, MY, YO, HO, KA, SY, Y Sugawara, and SM are employees of Preferred Networks, Inc. (Tokyo, Japan). HM is a contractor of Kao Corporation and a director of Preferred Networks, Inc.

*models*, make this challenging goal possible. We built a model that captures complex statistical interactions among more than 2000 human attributes and made it available as a software service on the Internet. The model can be used for estimating unknown attributes from known attributes and generating synthetic data. We believe that this model significantly lowers the barrier to entry into digital healthcare and will stimulate future innovations.

## Introduction

Advances in information technology, particularly in machine-learning and sensing technologies, are revolutionizing human healthcare by enabling continuous monitoring and analysis of individual health status [1–5]. Digital data collected through various measurements can be integrated to create comprehensive health profiles, allowing for early detection of health risks and personalized interventions. This data-driven approach is crucial for realizing predictive, preventive, and personalized medicine [6,7]. For example, by analyzing patterns in lifestyle, diet, and physical activity data, healthcare providers can develop tailored interventions that address individual risk factors before they lead to serious health conditions.

A major challenge in developing such healthcare solutions, however, is the limited availability of comprehensive health data. Data collection is often hindered by high costs, privacy concerns, and the fragmented nature of health records. Traditional approaches require collecting data specific for each application, making it inefficient and sometimes impractical to develop multiple healthcare solutions.

To address these challenges, we propose the Virtual Human Generative Model (VHGM), a novel statistical framework that can estimate missing attribute values from known attribute values and generate synthetic but realistic human health data. Unlike conventional statistical models, the VHGM can:

1. Capture complex relationships among over 2000 diverse health attributes
2. Estimate missing attribute values from known attribute values
3. Generate synthetic data that preserves the statistical properties of real populations
4. Support multiple healthcare applications through a single model

There are two key innovations in our approach. One is the integration of multiple independent data sources to create a high-dimensional training dataset. While existing clinical studies typically include fewer than 100 attributes [8], limiting their applicability, our method combines data from many sources using statistical linking techniques, without relying on personally identifiable information.

The other is maintaining the quality of the VHGM. Current deep learning-based machine learning is stochastic and there is no guarantee of the "correctness" of the model outputs. Among several ongoing technical discussions related to improving the trustworthiness of a model, some focus on augmenting the training datasets [9] and others focus on the characteristics of the deep neural network [10]. We take a practical approach to the VHGM, combining a robust set of quality metrics to objectively measure the model quality with a transparent governing process with multiple stakeholders.

The primary contributions of this paper are:

1. Development of a novel method for combining data from multiple clinical studies while preserving their statistical relationships

2. Implementation of the VHGM, a generative model with more than 2000 heterogeneous health attributes across diverse categories, together with a practical quality assurance process

3. Demonstration of the VHGM's practical utility through multiple real-world healthcare applications

## Materials and methods

### Design of the VHGM

The VHGM is produced and operated by three data-processing steps and one governing process (see Fig 1). Step 1 is to prepare the data. We used three data sources, each of which consists of one or more table-structured datasets. Data Source A was specifically prepared for the VHGM to obtain diverse health attributes simultaneously from approximately 1000 participants. Data Source B was commercially available data on annual health checkups and health insurance receipts of over one million individuals. Re-purposing of these data conforms to the Japanese Privacy Law and was approved by the Information Security Committee of Kao Corporation. Data Source C was a collection of previously reported studies to supplement specific health attributes. These studies were conducted internally by Kao Corporation (Tokyo, Japan), each of which was individually approved for that particular study. Re-purposing these datasets for the VHGM is covered by an umbrella approval in April 2021 by the IRB of the Kao Corporation and the Preferred Networks, Inc (Tokyo, Japan).

Step 2 is to integrate all the data sources into a single training dataset. The attributes to be extracted from each data source were determined by the *model schema*. The model schema also determines the data type of each attribute.

Step 3 is to train the generative model [11]. The resulting VHGM model was deployed as a commercial Application Programming Interface (API) service.

During the whole process, the governing committee oversaw the quality control of the VHGM.

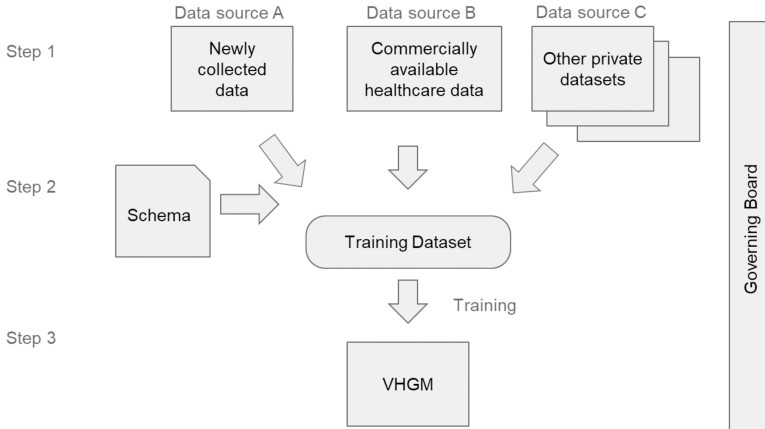

**Fig 1. Schematic diagram of the VHGM development and operation.** Data obtained from Data Sources A, B, and C are combined into a single training dataset to build a general-purpose statistical model called the VHGM, which is accessible via the API.

## Data Source A

Data Source A is from a single-center cross-sectional observational study that was conducted with adult men and women living in narrowly defined metropolitan areas of Japan (i.e., Tokyo, Kanagawa, Chiba, Saitama, Ibaraki, Tochigi, and Gunma prefectures). All measurements were performed by trained research coordinators and medical doctors using standard operating procedures during two outpatient visits to the Ueno Asagao Clinic (Tokyo, Japan), one week apart. The Strengthening the Reporting of Observational Studies in Epidemiology (STROBE) guidelines were applied according to the study objectives [12]. The study protocol is available online [13].

**Ethics approval, informed consent, and participation.** The study was approved in October 2021 by the IRB of Kao Corporation (Tokyo, Japan; approval #K0023-2108) and Preferred Networks, Inc (Tokyo, Japan; approval #ET22110047). Eligibility was evaluated by asking potential participants a few questions. All participants provided written informed consent to participate in the study. The consent form explained in detail which data would be used in the study and obtained consent for the use of anonymized data. It also stated that statistical models developed through the use of participants' anonymized data may be used in the future by Kao Corporation or its commissioned contractors. The study in Data Source A was registered at the University Hospital Medical Information Network (UMIN; UMIN000045746) on October 14, 2021. Recruitment started on October 19, 2021, and ended on February 25, 2022.

**Participants and eligibility.** Eligible participants were consecutively recruited over a 5-month period from October 2021 through February 2022. The participants were recruited via a website administered by TES Holdings (Tokyo, Japan). Participants were stratified into age groups by decade (20-29, 30-39, 40-49, 50-59, 60-69, and ≥ 70 years) to match the decade ratio of the typical adult Japanese population. The major inclusion criteria were as follows: (1) Japanese men and women aged ≥ 20 years and (2) individuals able to complete the questionnaires and surveys. Major exclusion criteria were as follows: (1) individuals undergoing hospitalization for serious diseases (e.g., diabetes, hypertension, arteriosclerosis, heart disease, malignancy, Alzheimer's disease, etc.), (2) individuals who could not come to the outpatient unit by themselves, and (3) individuals with dementia or suspected dementia. The detailed inclusion and exclusion criteria are available in the protocol [13].

**Measurements and data processing.** Numerous health attributes across diverse categories were collected. The parameters were grouped into the following 16 measurement categories: blood pressure and arterial stiffness, lifestyle investigation and questionnaire, cognitive function analysis, laboratory analysis, oral glucose tolerance test, anthropometric measurements, skin surface spectroscopy, physical performance tests, hand surface analysis, liquid chromatography-tandem mass spectrometry, body odor analysis, lipids in the stratum corneum and sebum analysis, hair loss determination, lipid mediator analysis, skin surface lipids (SSL)-RNA analysis, and microbiota analysis, based on the measurement methods described in the protocol [13]. Details on the SSL-RNA, intestinal microbiota, and saliva microbiota are described in S1 Text. The data management details are also described in the protocol paper [13].

**Data analysis.** We reviewed basic statistical characteristics of the data and compared them with recent official statistics in Japan. A correlation matrix using the Spearman rank correlation was generated to examine relationships between attributes and data sparseness. To create the correlation matrix, only real, positive, ordered categorical, and binary attributes (i.e., categorical attributes with only two possible values) were used.

## Data Source B

In Japan, all employers are required to provide healthcare insurance coverage for their employees through company-specific insurance associations. Under legislation enacted by the Japanese government, healthcare-related records can be utilized for research and development purposes without individual consent, provided they undergo an approved anonymization process [14]. Several private-sector data aggregators make such anonymized data commercially available. Data Source B consists of two comprehensive datasets covering the same set of approximately one million individuals, including both employees and their dependents in Japan in 2019:

1. Annual health examination records, including:
   - Physical measurements (height, weight, etc.)
   - Blood test results
   - Responses to questionnaires about health-related lifestyle factors
2. Medical and dental consultation records, including:
   - Diagnosed conditions
   - Diagnostic tests and treatments performed
   - Drug prescriptions administered
   - Insurance points (used for calculating reimbursement amounts)

We preprocessed the purchased data, selecting 56 attributes from the annual health examination records and extracting 199 attributes for major disease diagnosed, major test and treatment procedures performed, or major drugs administered, each of which represents how many times the individual visited the doctor for that particular disease, procedure, or prescription drug during the year. In addition, we also added three quantitative attributes: one for the total insurance points of the year (roughly representing how much money the individual spent on medical services in the year), one for the insurance points related to medical (non-dental) services, and one for the insurance points related to dental services. Furthermore, we introduced three binary flag attributes representing service utilization: one indicating whether the individual ever visited a medical (non-dental) doctor in the year, one indicating whether the individual ever visited a dentist in the year, and one indicating whether the individual used both medical and dental services in the year.

## Data Source C

The collection criteria of previous studies in Kao Corporation were as follows: 1. an adequate number of participants for modeling ($\geq$ 100 participants in each clinical study), 2. inclusion of common attributes (age, sex, height, weight, etc.), and 3. gender balance (i.e., exclusion of datasets containing only male or only female participants). Under these criteria, we selected one cross-sectional study on visceral fat accumulation [15] as Data Source C-1 and 12 intervention clinical trials on drinks including green tea catechins and coffee chlorogenic acids [16–27] as Data Source C-2. The cross-sectional study of Data Source C-1 includes basic measurements such as general blood testing and lifestyle questionnaires. The intervention trials of Data Source C-2 include basic measurements (e.g., weight, height, and blood pressure) and specialized measurements (e.g., visceral fat area, lipid profile, and gastrointestinal hormones) to assess metabolic syndromes. Data from the first visit before interventions were extracted, and the subsequent changes observed after exposure to the active ingredients were added to represent participant baseline characteristics, captureing both their initial status and responsiveness at a single reference time point.

## Design of model schema

The VHGM represents a joint probability distribution over a set of random variables $X_1, X_2, ..., X_k$, where each random variable represents an *attribute* of human health data. The *model schema* defines which attributes to extract from the data sources, along with their domains and semantic interpretations. While maximizing the number of attributes could potentially increase the model's utility for future applications, including superfluous attributes that are rarely used in downstream tasks or contribute minimally to the estimation of other attributes can adversely affect both model accuracy and computational efficiency. Therefore, we established the following criteria for attribute selection:

1. Missing rate: Attributes with high missing data rates are excluded as they lead to less reliable estimations
2. Potential utility: Attributes with limited applicability in anticipated future applications are omitted
3. Independence: Attributes showing minimal correlation with other variables are less valuable for joint estimation and may be excluded

The model is designed to handle *heterogeneous* data types, accommodating various attribute distributions and domains. We assume that each type has a parametric distribution, such as the Gaussian distribution. Based on our training algorithm requirements [11], we categorized attributes into the following types:

1. *Real*: Continuous variables following a normal distribution (e.g., height)
2. *Positive*: Strictly positive continuous variables following a log-normal distribution (e.g., blood glucose levels)
3. *Count*: Discrete variables following a Poisson distribution (e.g., number of doctor visits)
4. *Categorical*: Nominal variables with a finite set of unordered options (e.g., sex)
5. *Ordered Categorical*: Categorical variables with inherent ordering (e.g., drinking habit as in *Never, Sometimes, Everyday*)

## Model algorithm

**Combining multiple datasets.** Privacy-Preserving Record Linkage Systems are tools designed to link records across multiple datasets while protecting individual privacy [28]. These systems typically require access to personally identifiable information prior to the de-identification process and assume the existence of a sufficient number of common subjects across datasets. The VHGM training algorithm [11] takes a fundamentally different approach, eliminating the need for personally identifiable information or common subjects across datasets. Instead, it employs statistical linkage, leveraging common attributes (such as age and sex) that naturally occur across different datasets without requiring shared identifiers.

The statistical linkage in the VHGM can be conceptually expressed through transitional conditional probability (implementation details are given in [11]). To establish an "indirect relation" between variable $X$ in dataset 1 and variable $Y$ in dataset 2, one may:

1. Estimate the conditional probability distribution $P(Z|X)$ in dataset 1
2. Estimate the conditional probability distribution $P(Y|Z)$ in dataset 2
3. Apply the marginalization rule of conditional probability to calculate

$$P(Y|X) = \int P(Y|Z,X)P(Z|X)dz$$

where $Z$ represents common variables present in both datasets. Here, we assume $P(Y|Z) = P(Y|Z, X)$, i.e., given $Z$, $Y$ is independent of $X$.

The process of combining these heterogeneous datasets into a unified training dataset is illustrated in Fig 2. We employed a row-wise concatenation approach, where:

- Each record from the source datasets becomes a separate record in the combined dataset
- Attributes not present in a particular source dataset are treated as missing values
- Common attributes across datasets (e.g., age and sex) serve as implicit linking features

**Model architecture.** Given the systematic nature of missing values in our combined dataset (as opposed to random missingness), we designed a model architecture that is robust in the presence of large-scale non-random missingness patterns [11]. The main idea is inspired by Vision Transformer (ViT) [29] for image recognition where an input image is split into a set of image "patches," and a Transformer is used to capture the semantic relationships among these patches.

Similarly, instead of treating the input as a fixed-size vector in many table-based machine learning systems, our algorithm takes a sequence of "tokens" corresponding to observed attributes as if it were a sequence of words. These tokens are embedded into a fixed-dimensional space, accommodating various encoding schemes such as one-hot encoding for categorical attributes.

Fig 3 illustrates the architecture of our model. Our transformer-based encoder leverages attention mechanisms to capture the relationships among the observed input tokens and transforms it into a sequence of latent representations (blue boxes in the diagram). Missing attributes do not contribute during this encoding process. Subsequently, the latent representation for all attributes is constructed by combining the transformed tokens for observed attributes and a default learnable token assigned to each missing attribute (yellow boxes in the diagram). From this unified latent representation, the decoder generates the estimated distribution of every attribute. The output of each attribute is a set of estimated parameters for the attribute, depending on its type (e.g., mean and standard deviation for *real* attribute modeled as Gaussian).

**Model training.** We employed two techniques to train the model. The first is Masked Modeling [30], in which certain input attributes are intentionally masked, and the model is trained to recreate their original values. This approach is particularly effective for the missing-value imputation task targeted by the VHGM.

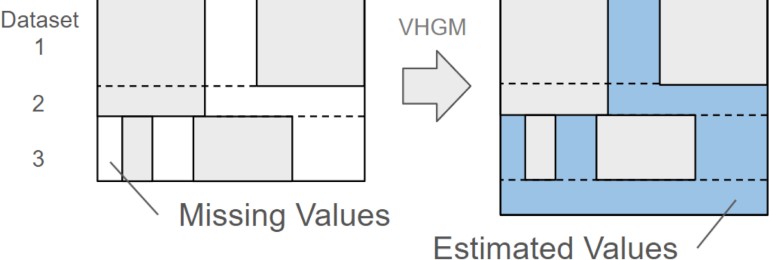

**Fig 2. Dataset concatenation and data generation.** The diagram shows the relationship between records (rows) and attributes (columns), with distinct datasets represented as row blocks. White areas indicate missing values and blue areas represent estimated values. Additional rows on the right-hand side indicate synthetic records generated by the model.

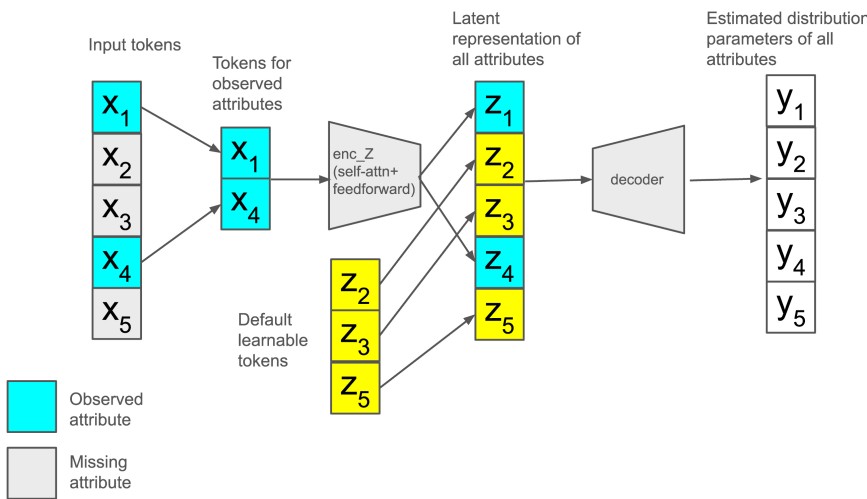

**Fig 3. Model architecture.** Tokens for observed attributes (blue) are transformed into latent representations and combined with a default learnable token for each missing attributes (yellow).

The other is a *two-stage training approach*: In the first stage, the model is trained separately on individual datasets while in the second stage the model is fine-tuned by the combined dataset. We found that this two-stage training strategy significantly reduced training time without noticeable degradation in accuracy.

**Model quality.** The VHGM aims to approximate the joint probability distribution across more than 2000 attributes. Therefore, its quality should be evaluated based on how accurately it captures the underlying real-world distribution. This evaluation presents three significant technical challenges:

1. **The challenge of ground truth:** The actual real-world distribution is unknown and cannot be precisely estimated from observed data. To address this, we employed the standard approach of data splitting:
   - For each of the data sources, randomly select 10% of the original records as a holdout set, maintaining the sex and age distributions
   - Use the remaining 90% for model training
   - Maintain strict isolation of the holdout set from model developers to prevent information leakage and ensure unbiased evaluation
   - Evaluate imputation errors using the holdout set

   Using the 10% holdout sets, we calculated the imputation errors for each attribute in each data source. The value of a target attribute in a record from the holdout set was masked, and an imputed value was obtained using 10 attributes in the same record as input to the VHGM. These 10 attributes were chosen based on their strong relationship with the target attribute using the VHGM. Errors were calculated by comparing the target attribute value with the imputed value. The method details, depending on the data type of the target attribute, are described in S2 Text.

2. **The challenge of high dimensionality:** Evaluating the similarity between high-dimensional joint distributions is inherently complex. In the absence of standardized methods for such evaluation, we developed a practical three-component assessment framework:

i. *Univariate analysis:* For each attribute, we compared the marginal distributions between the training dataset and model output. For attributes of type *real*, we evaluated the overlapping area between the histograms of the training dataset and model output. Detailed methods are described in S3 Text.

ii. *Bivariate analysis:* For 70 pre-selected attribute pairs $\langle X, Y \rangle$ that exhibited high correlations in the training dataset, we compared the conditional distributions $P(Y|X)$ between the training data and model output. Detailed methods are described in S3 Text.

iii. *Scenario-based analysis:* In this study, "scenario-based analysis" refers to an evaluation approach in which the VHGM is tested under predefined conditions that represent realistic or hypothetical use cases relevant to potential applications. In this analysis, outputs were observed in the model response to pre-selected inputs based on anticipated use case scenarios. Given that there may not always be sufficient records in the training data to validate the same combinations of input values, we assessed the direction and magnitude of changes in the obtained results to ensure they were intuitively consistent and comparable to prior knowledge.

3. **The challenge of validation with external datasets:** Validating a generative model against external datasets poses substantial challenges due to variations in data collection protocols and population demographics. To assess the model's generalizability, we conducted a comparative analysis using two well-established, independent datasets: the National Health and Nutrition Survey, available via the Portal Site of Official Statistics of Japan (e-Stat) [31], and the U.S. National Health and Nutrition Examination Survey (NHANES) [32].

   The validation focused on 24 nutritional intake attributes (e.g., calories, protein) that were derived from diet record or recall methods and determined to exhibit high semantic equivalence across all three datasets (see S4 Text). For each attribute, we statistically compared the distributions to evaluate concordance. Specifically, we treated the mean value from the VHGM output as a sample mean and calculated its z-score relative to the corresponding distribution in each external dataset (e-Stat or NHANES). Assuming normality, we further computed the overlapping area between the VHGM distribution and each external dataset, providing a quantitative measure of distributional similarity.

In addition to these evaluation components, we also conducted a benchmark comparison with widely used tabular generative models, including TVAE, CTGAN [33], and Gaussian Copula. As a common performance metric for this comparison, we adopted *Machine Learning Efficiency*, which evaluates whether synthetic data can give rise to a machine learning model with performance comparable to that trained on the original data. Because not all baseline models can handle discrete variables, we employed a regression task.

This evaluation approach aligns with similar metrics used in recent synthetic healthcare data generation efforts [8]. Previous studies similarly emphasized the importance of measuring the fidelity of synthetic data across multiple dimensions of analysis.

## Governance process

In this study, the term "governance process" refers to the structured set of policies, decision-making mechanisms, and oversight activities that define stakeholder roles, guide schema updates and new model releases, and manage risks to ensure the secure, ethical, and effective operation of the VHGM. Our governance framework addresses the needs and concerns of four key stakeholder groups:

1. **Data subjects**: Individuals whose health data contribute to the model. Their primary concerns are data privacy and protection, which are mitigated through robust anonymization protocols.
2. **Data owners**: Organizations providing source datasets. Their primary concerns are data security and protection of intellectual property, which are mitigated through contractual agreements, system security, and usage monitoring.
3. **Application owners**: Organizations providing healthcare solutions using the VHGM. Their primary concerns are model reliability, availability, and performance, which are supported by technical documentation and service-level agreements.
4. **End users**: Consumers of VHGM-based applications. Their primary concerns are trustworthiness and validity of the VHGM outputs, which are addressed through transparent validation processes and a clear limitations disclosure.

To ensure effective oversight, we established a multi-stakeholder governance committee that (1) conducts monthly meetings to review operations, (2) makes final decisions on the model schema, (3) approves new model releases, and (4) evaluates potential risks and determines mitigation strategies. Additionally, the committee members are carefully nominated to cover diverse backgrounds, including life science researchers, clinical study experts, data scientists, and marketers with expertise in healthcare applications, thereby ensuring that users' perspectives, healthcare demands, and ethical and privacy matters are appropriately considered. The terms and conditions of the VHGM API service are also carefully designed to ensure governance in providing healthcare solutions using the VHGM.

During this whole process, we maintained transparency through open communication with multiple channels as follows:

- Publication of technical papers on study planning [13], outcomes (this paper), and training algorithms [11]
- Monthly newsletters [35]
- Clear documentation of model capabilities and limitations

Machine learning models can never be perfect and require continuous refinement. As such, we periodically release newer versions of the VHGM model. This process is often referred to as "MLOps" [9], and is known to be complex because improvements in certain aspects may affect others. Through the transparent governance process described above, the VHGM allows stakeholders to make informed decisions about which model version best suits their specific needs.

## Results

This section describes the results for the latest model of the VHGM as of August 2025, named *pollux* (see Table 11).

### Data Source A

A total of 997 participants were included in the study and their data were obtained. Three of the participants placed restrictions on the use of their data and did not consent to secondary use of their data. Thus, the data from 994 participants were used for data characterization, model development, and model applications. Fig 4 summarizes the participants' flow process.

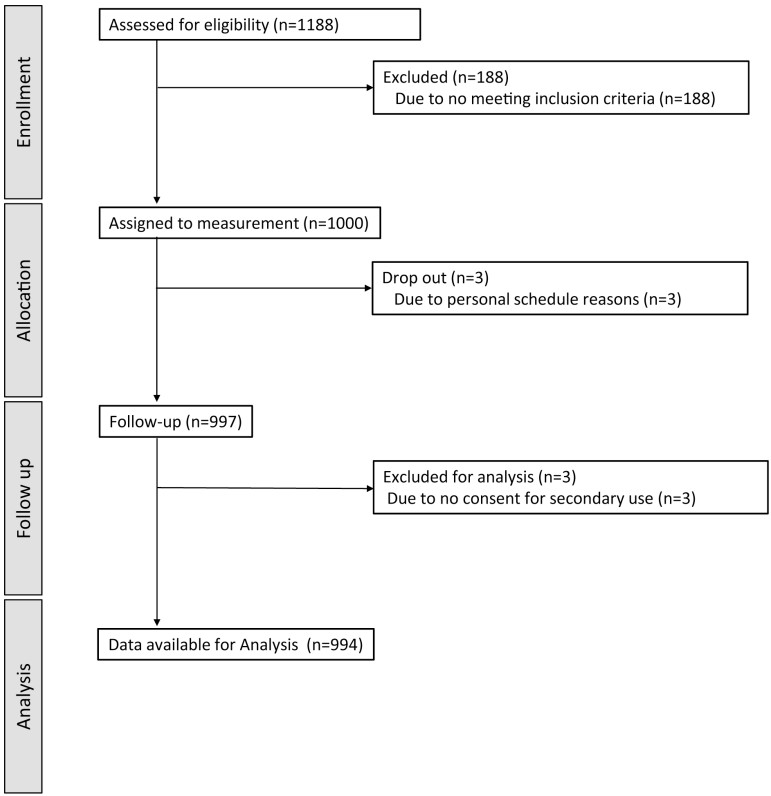

**Fig 4. The participants' flow of Data Source A.** The number of participants in the entire recruitment process.

**Participants.** The sex and age ratios of participants in these visits closely mirror those of the adult Japanese population (see the details in S1 Table and S2 Table). Table 1 shows the characteristics of the participants. These values are consistent with recent official statistics provided by the Japanese government [36]. Thus, this dataset may approximately represent the statistical characteristics of the Japanese population.

**Data analysis.** The preprocessed data under the latest model schema includes a total of 1868 attributes that came from the 16 measurement categories. This number is comparable to a large cross-sectional study [37]. Attribute number, missing rate, and outlier rate of each measurement category are shown in S4 Table. Except for Hair Loss Determination and Lipid Mediator Detection, which were relatively difficult to measure, the missing data rate in the other measurement categories was less than 20%, indicating a sufficiently low rate of missing

**Table 1. Data Source A participant characteristics.**

| Characteristics | Male | Female | Male (Ref.) | Female (Ref.) |
|---|---|---|---|---|
| Age (y), mean (SD) | 51.36 (16.27) | 51.16 (16.51) | n/a | n/a |
| Height (cm), mean (SD) | 169.80 (5.83) | 157.19 (5.74) | 167.7 (6.9) | 154.3 (6.7) |
| Weight (kg), mean (SD) | 68.88 (12.42) | 52.79 (9.76) | 67.4 (12.0) | 53.6 (9.2) |
| BMI (kg/m$^2$), mean (SD) | 23.87 (3.96) | 21.35 (3.69) | 23.9 (3.6) | 22.5(3.7) |

Mean (standard deviation [SD]) of height, weight, and BMI were calculated using only adults ($\geq$ 20 years old). The Male (Ref.) and Female (Ref.) data are available at e-Stat [36].

data. Some measurement categories exhibited higher outlier rates than others. Most of these abnormal values were thought to be caused by diseases such as diabetes rather than by noise or other factors, and therefore the data quality was considered to be high. Fig 5 shows the correlation matrix for Data Source A. Many strong relationships existed between variables in the same measurement category, but there were some weak relationships between variables in different measurement categories. This dataset had a sparse data structure, as many pairs of variables had no or weak relationships (see S1 Fig). Many strong relationships between pairs of variables were observed not only within the same measurement category but also across different measurement categories (see several examples in S2 Fig).

## Data Source B

The record and attribute numbers of the preprocessed data were 1,245,807 and 261, respectively, under the latest model schema. The record number corresponds to approximately 1% of adults living in Japan. This dataset is expected to adequately reflect the statistical characteristics of the annual health checkup and the medical and dental receipts, although the older adult population ($\geq 60$) was lower due to their retirement and the ratio of males to females was slightly higher due to several possible reasons (e.g., employment, income, lifestyle, etc.) [38]. The missing rates were considerably low. Almost all these outliers were thought to be caused by diseases, and thus the data quality was considered to be sufficient.

## Data Source C

The record and attribute numbers of the data of Data Source C-1 were 11,646 and 61, respectively, under the latest model schema. The missing rate was considerably low due to the high quality control of the study. Because the study was conducted mainly at workplaces, the ratio

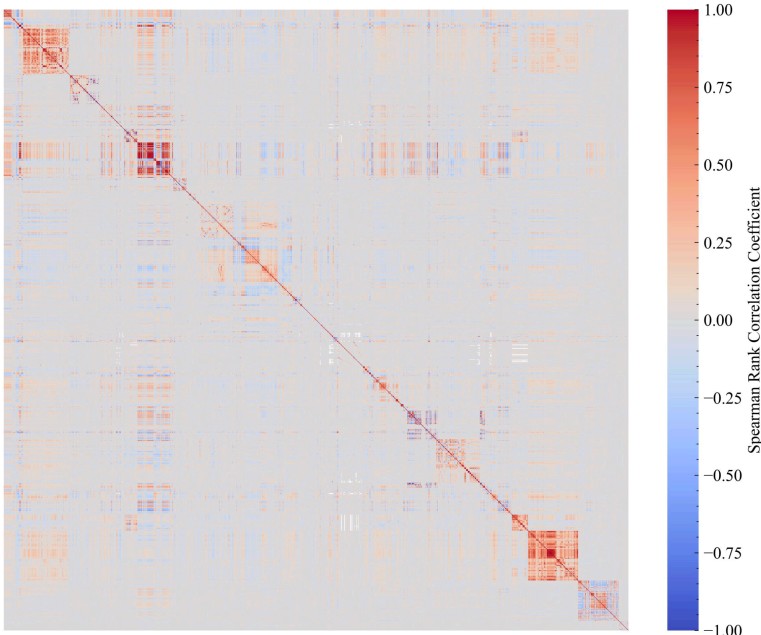

**Fig 5. Correlation matrix using the Spearman rank correlation.** The color represents the correlation coefficient between real, positive, ordered categorical, and binary categorical attributes.

of males to females was high. The record and attribute numbers of the aggregated data of Data Source C-2 were 1745 and 162, respectively. The distributions of age, sex, and BMI were not perfectly matched with the overall population in Japan due to the inclusion and exclusion criteria of the intervention studies.

## Model schema and training dataset

Table 2 shows the numbers of the preprocessed dataset records. The original datasets were split into 90% for the training dataset and 10% for the holdout dataset; the training dataset was then selected and augmented to incorporate four imbalanced records from various data sources, as illustrated in Table 2. The attribute overlaps from the Data Sources A, B, C-1, and C-2 are described in Fig 6. Table 3 shows attribute occurrence percentages in Data Sources A, B, C-1, and C-2. As shown in Table 3, common attributes (age, sex, weight, height, BMI, etc.) were used to connect all the Data Sources. The number of each type of attribute for each of the data sources is provided in Table 4. As clearly shown in Table 5, this dataset encompasses a range of field categories, defined by the authors based on application fields, from "Vital signs" to "Lifestyle" enabled by multiple data sources. This diversity is attributed to the multiple data sources, particularly Data Source A, which includes various health attributes. Not only "Demographic", but also "General blood testing" functioned as "common" attributes.

## VHGM quality

**Missing value imputation.** The imputation performance was evaluated using the holdout set, which corresponds to 10% of the original datasets. Since the model output for each attribute is a distribution, we calculated the errors by treating the *mode* of the estimated distribution as if it is the point estimation. The means (standard deviations) of standardized

**Table 2. Numbers of records of each data source.**

| Data Source | Original records | 90% Records | Adjusted records |
|---|---|---|---|
| A | 994 | 897 | 18,000 (up sampling) |
| B | 1,245,807 | 1,121,227 | 100,000 (down sampling) |
| C-1 | 11,646 | 10,483 | 18,000 (up sampling) |
| C-2 | 1745 | 1584 | 18,000 (down sampling) |

This table shows the number of records in Data Sources A, B, C-1, and C-2. 90% Records corresponds to the dataset without the holdout set. Adjusted records correspond to the dataset that was selected and augmented for training the model.

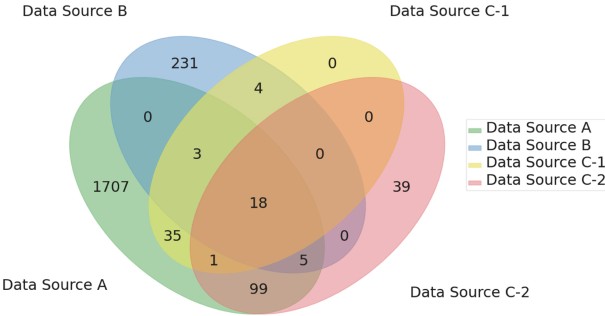

**Fig 6. Overlap of attributes across datasets.** This image shows attribute overlaps across the Data Sources A, B, C-1, and C-2.

**Table 3**. **Attribute occurrence across Data Sources.**

| Occurrence | Attribute number (%) |
|---|---|
| 1 | 1977 (92.2) |
| 2 | 138 (6.4) |
| 3 | 9 (0.4) |
| 4 | 18 (0.8) |

This table shows the attribute occurrence percentages in Data Sources A, B, C-1, and C-2.

**Table 4**. **Number of attributes of each data type for the data sources.**

| Attribute Type | Data Source A | Data Source B | Data Source C-1 | Data Source C-2 |
|---|---|---|---|---|
| Real | 1134 | 15 | 11 | 145 |
| Positive | 15 | 10 | 6 | 7 |
| Count | 0 | 199 | 0 | 0 |
| Categorical | 485 | 27 | 5 | 2 |
| Ordered Categorical | 234 | 10 | 39 | 8 |
| Total | 1868 | 261 | 61 | 162 |

The data type definition is described in Design of model schema.

**Table 5**. **The field categories of the attributes in each data source.**

| Field | Source A | Source B | Source C-1 | Source C-2 |
|---|---|---|---|---|
| Vascular function | 53 | n/a | n/a | 4 |
| Nutrient | 243 | n/a | n/a | 63 |
| Cognitive function | 69 | n/a | n/a | n/a |
| Liver function | 16 | 3 | 3 | 11 |
| Renal function | 17 | 3 | 1 | 14 |
| General blood testing | 20 | 7 | 4 | 18 |
| Sugar metabolism function | 61 | 7 | 3 | 10 |
| Hormone | 10 | n/a | n/a | n/a |
| Stress / Fatigue | 49 | n/a | n/a | n/a |
| Body composition / Physique | 77 | 7 | 5 | 15 |
| Vital signs | 4 | 2 | 2 | 6 |
| Women's health | 213 | n/a | n/a | 1 |
| Skin care / Hair care | 193 | n/a | n/a | n/a |
| Motor function | 20 | 2 | n/a | n/a |
| Immunity / Hygiene habit | 108 | n/a | n/a | n/a |
| Lifestyle | 57 | 10 | 38 | n/a |
| Sleep | 25 | n/a | n/a | n/a |
| Constitutional classification / Personality | 9 | n/a | n/a | n/a |
| Body odor | 28 | n/a | n/a | n/a |
| Excretion function | 27 | n/a | n/a | n/a |
| Health awareness | 76 | n/a | n/a | n/a |
| Oral cavity | 33 | 1 | n/a | n/a |
| Productivity / Presentism | 36 | n/a | n/a | n/a |
| Demographic | 4 | 2 | 2 | 2 |
| Medical history / Medication (questionnaire, number of receipts issued) | 55 | 189 | 3 | n/a |
| Nursing care | 1 | n/a | n/a | n/a |
| Biomarker | 181 | n/a | n/a | n/a |
| Walking characteristics | 183 | 1 | n/a | 1 |
| Medical procedure (number of receipts issued) | n/a | 27 | n/a | n/a |
| Other fields | n/a | n/a | n/a | 17 |
| Total | 1868 | 261 | 61 | 162 |

The field categories were defined and each attribute was grouped into a single field category by the authors the same way as for the measurement categories in Data Source A.

errors for real, positive, and count-type attributes were 0.527 (0.550), 0.566 (0.545), and 0.129 (0.589), respectively (Table 6). The means of accuracies for ordered categorical type attributes were lower than those for categorical type attributes (Table 7). This discrepancy is likely due to the greater number of selections for ordered categorical type attributes compared to categorical type attributes. These errors were better than those obtained using the mode imputation for the categorical type, the ordered categorical type, and the count type and the mean imputation for the real type and the positive type, following the same trend observed with the training dataset [11]. Thus, this method is practically acceptable for applications involving missing value imputations in setting comparable to our data.

**Univariate and bivariate analyses.** Fig 7 shows the result examples of univariate and bivariate analyses. With these metrics, we ensured that the model captured important statistical properties of the training dataset. Note that in Fig 7(b), non-linear relationships between attributes are properly captured, which would not be possible with a simple linear model such as the covariance matrix.

**Table 6. Imputation errors in real, positive, and count-type attributes.**

|  | Real | Positive | Count |
|---|---|---|---|
| Attribute number | 1170 | 18 | 199 |
| Absolute standardized error (Mean) | 0.527 | 0.566 | 0.129 |
| Absolute standardized error (Standard deviation) | 0.550 | 0.545 | 0.589 |
| The details are provided in S2 Text. | | | |

**Table 7. Imputation errors in categorical and ordered categorical type attributes.**

|  | Categorical | Ordered Categorical |
|---|---|---|
| Attribute number | 511 | 244 |
| Averaged Accuracy (Mean) | 0.839 | 0.573 |
| The details are provided in S2 Text. | | |

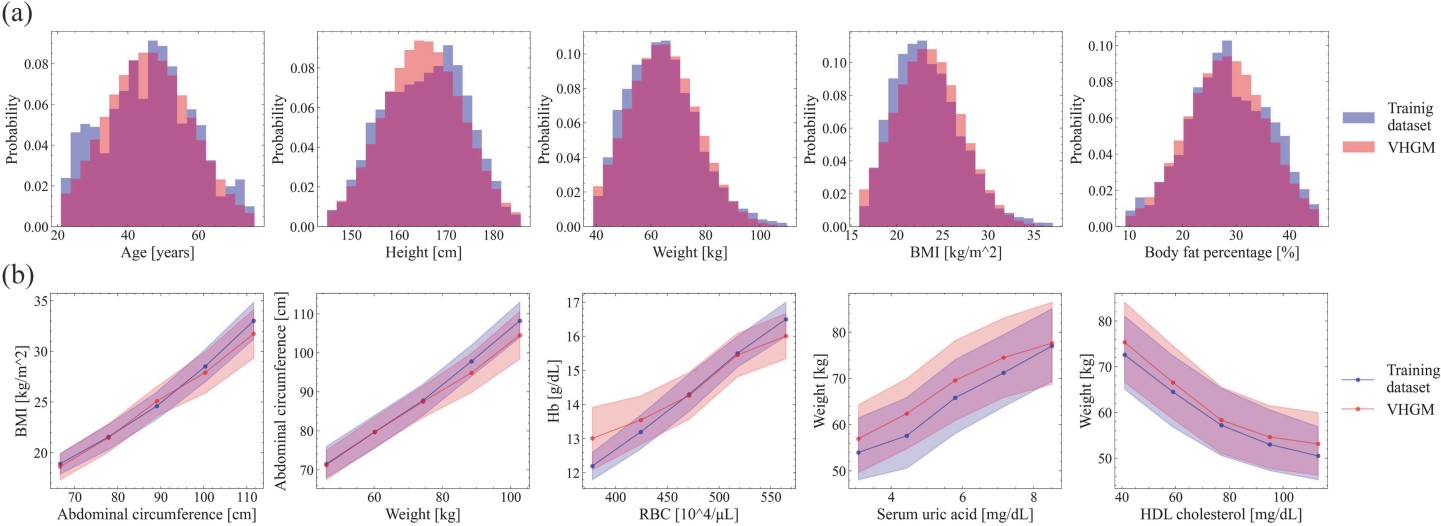

**Fig 7. Univariate and bivariate analyses.** (a) Univariate comparison between the training dataset and model. (b) Bivariate comparison between the training dataset and model. The blue in the graphs shows the distribution of the training dataset and the red shows the distribution of the model.

**Scenario-based analysis.** Fig 8 shows an example of scenario-based analysis. In this case, the input to the VHGM consists of five basic attributes (sex, age, weight, height, and BMI) and five lifestyle attributes (e.g., "I am proactive in eating green and yellow vegetables") for three different personas: *healthy*, *normal*, and *unhealthy*. The VHGM estimates the levels of nutrient intake for these personas.

As illustrated in the figure, a higher intake of carbohydrates was associated with unhealthy habits, while lower levels of dietary fiber and vitamin C intake were estimated for unhealthy personas. These results are generally consistent with common knowledge regarding the relationships between nutrient intake and lifestyle, and we interpret this as supporting evidence that the VHGM captures real-world patterns.

Of course, scenario-based analysis is not exhaustive, and in some cases, the VHGM may generate counterintuitive results. We found that scenario-based analysis can serve as a quick test for assessing model quality, without requiring computationally expensive imputation or univariate/bivariate analyses.

**External dataset validation.** The results of the external validation are summarized in Tables 8 and 9, which detail the z-scores and overlap areas from the comparison between the VHGM output and the e-Stat and NHANES datasets, respectively. The analysis indicates a higher degree of statistical consistency between the VHGM output and the e-Stat data (mean overlap area = 0.82) compared to the NHANES data (mean overlap area = 0.59). This finding is expected, as the model's training data was sourced from a Japanese population, which is more demographically similar to the population represented in e-Stat than to the US-based population in NHANES.

## (a) Input

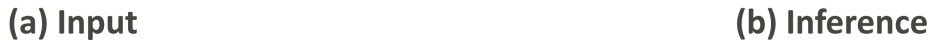

| Attribute | Healthy habit | Normal habit | Unhealthy habit |
|---|---|---|---|
| **Basic information** | | | |
| Sex | M | M | M |
| Age | 45 | 45 | 45 |
| Weight | 70 | 70 | 70 |
| Height | 170 | 170 | 170 |
| BMI | 24.2 | 24.2 | 24.2 |
| **Lifestyle questionnaire** | | | |
| I am proactive in eating green and yellow vegetables | applicable | neither | not applicable |
| I like hamburgers, donuts, and potato chips | not applicable | neither | applicable |
| I try to hold off on animal fats and instead get plant-based or fish fats | Applicable | neither | not applicable |
| My mealtimes are irregular | not applicable | neither | applicable |
| I think I am more out of shape than overeating | not applicable | neither | applicable |

## (b) Inference

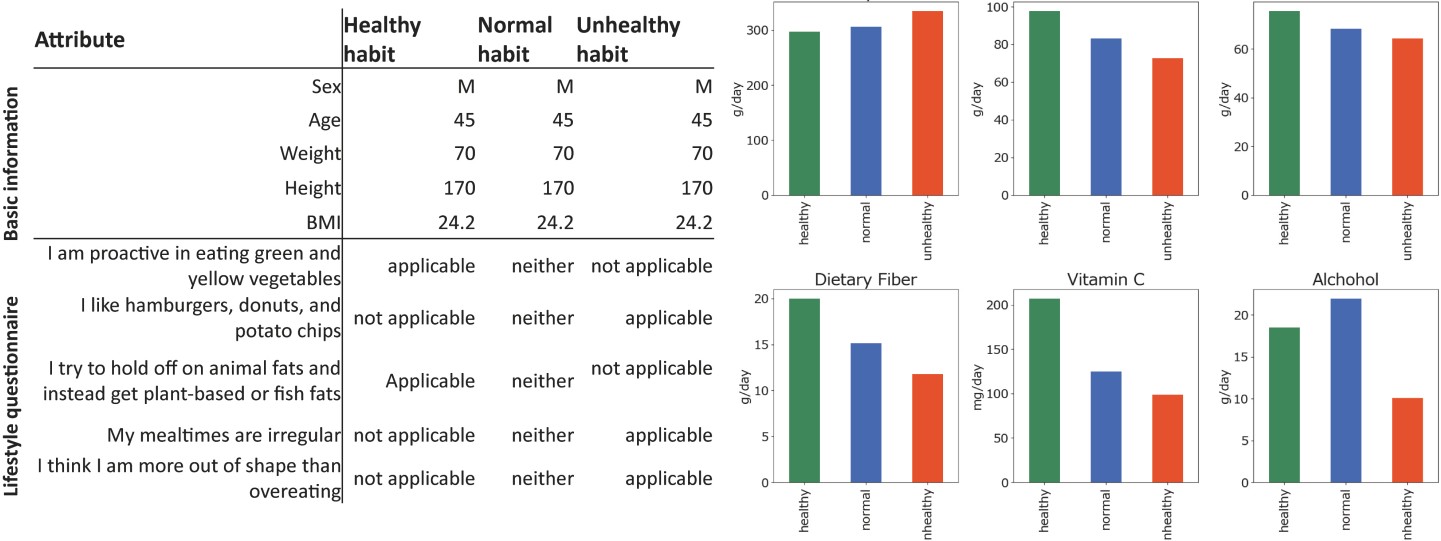

**Fig 8. Scenario-based analysis.** (a) Three personas (*healthy*, *normal*, and *unhealthy*) with five common attributes and five varying attributes on lifestyles. (b) Estimated nutrient intake per day for the personas.

**Table 8**. **Overview of the statistical difference between VHGM and e-Stats.**

| Metric | Mean | Std | Max | Min |
|---|---|---|---|---|
| Abs. z-score | 0.15 | 0.08 | 0.28 | 0.01 |
| Overlap area | 0.82 | 0.05 | 0.90 | 0.70 |

The table shows summary statistics for the 24 common attributes. The absolute (abs.) z-score was calculated by treating the mean from the VHGM output as a sample mean relative to the distribution of the corresponding attribute in the e-Stat dataset. The overlap area quantifies the similarity between the two distributions.

**Table 9**. **Overview of the statistical difference between VHGM and NHANES.**

| Metric | Mean | Std | Max | Min |
|---|---|---|---|---|
| Abs. z-score | 0.38 | 0.23 | 0.82 | 0.01 |
| Overlap area | 0.59 | 0.15 | 0.84 | 0.35 |

Absolute z-scores and overlap areas were evaluated using the same methodology as in Table 8, with NHANES as the external comparison dataset.

**Comparison with other generative models.** Table 10 shows the results of a regression task for TVAE, CTGAN [33], and Gaussian Copula along with our algorithm [11]. Our algorithm outperforms the known generative models for the synthetic data (the "Individual" column in the table). Interestingly, we can obtain better performance if the synthetic data is used for augmenting the original data (the second column in the table).

## Governance process

Table 11 summarizes the models published as of August 2025, based on the decisions of the multi-stakeholder committee. The number of attributes has generally increased with addition of new clinical data into the data sources and introduction of new attributes in the model schema. Model inference performance has also improved due to incremental algorithm updates [11]. As shown in Fig 9, model inference performance in bivariate analysis has improved.

Although the training dataset of the VHGM does not contain any personally identifiable information, the improvement of model inference performance may increase potential privacy concerns regarding *membership inference attacks*, which aim to determine if a particular individual was included in the training dataset. As recommended by the committee, we conducted a preliminary assessment of vulnerability to membership inference attacks and

**Table 10**. **Comparison with other algorithms on a regression task.**

| Dataset | Individual ($R^2$) | Combined with original ($R^2$) |
|---|---|---|
| Original | 0.531 | - |
| TVAE | 0.225 | 0.525 |
| CTGAN | -0.130 | 0.506 |
| Gaussian Copula | 0.476 | 0.531 |
| Ours | **0.501** | **0.553** |

Machine Learning Efficiency: the task is to estimate the depression score from 65 attributes related to exercise, nutrients, sleeping habits, stress, and tiredness. We generated 10,000 synthetic records for the 66 attributes. $R^2$ denotes the coefficient of determination.

**Table 11**. **Published models as an API service and their model parameter sizes.**

| Model Name | Date of Publish | Feature | Attributes | Parameter Size |
|---|---|---|---|---|
| *fomalhaut* | Dec. 7, 2022 | General-purpose | 1841 | 1,668,232 |
| *helvetios-a* | Feb. 2, 2023 | Health checkup focus | 260 | 290,266,849 |
| *helvetios-b* | Feb. 2, 2023 | Health checkup focus | 302 | 11,056,686 |
| *intercrus* | May 12, 2023 | General-purpose | 1975 | 85,822,679 |
| *libertas* | June 30, 2023 | General-purpose | 2150 | 94,131,094 |
| *lich* | Oct. 6, 2023 | General-purpose | 2097 | 26,951,129 |
| *musica* | April 8, 2024 | General-purpose | 2110 | 32,381,324 |
| *pollux* | May 21, 2025 | General-purpose | 2142 | 2,736,600 |

The model parameter size is typically dominated by the number of hidden nodes and the number of layers in the model architecture. The model names are based on the names of stars.

(a) fomalhaut

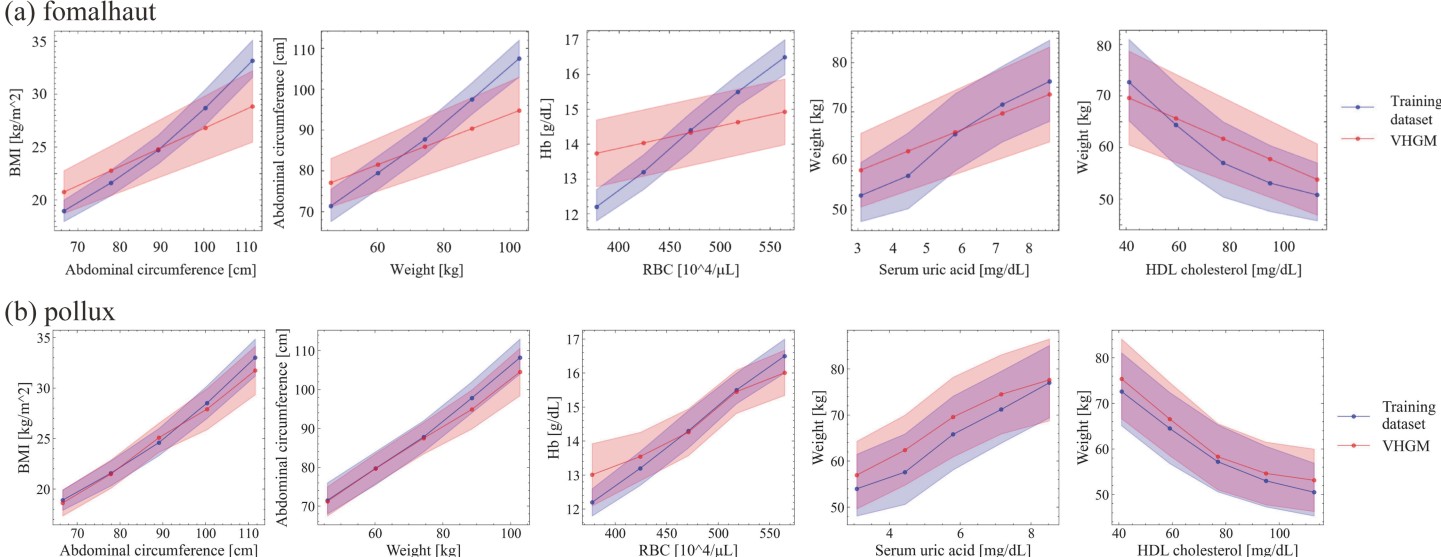

(b) pollux

**Fig 9. Improved model performance in bivariate relationships.** Overlap areas between the dataset and model output were increased for each attribute pair. (a) Bivariate relationships using *fomalhaut* published in Oct. 2022. (b) Bivariate relationships using *pollux* published in May 2025.

reviewed the results. A summary of these preliminary experiments has been reported elsewhere [34], indicating no immediate evidence of exploitable risk. Nevertheless, we acknowledge that a more comprehensive empirical evaluation is warranted and plan to address this in future work.

Estimated health information can potentially contribute to unethical decisions, such as ones that result in discrimination against underrepresented groups. The VHGM is a general-purpose API service and it is extremely difficult – if not impossible – to implement technical safeguards that fully prevent misuse. Instead, the terms and conditions of the VHGM explicitly prohibit antisocial or unethical uses of the technology. Furthermore, the committee also continuously monitors API usage to ensure compliance.

## Discussion

### Principal findings

The principal findings of this paper are twofold. First, a general-purpose generative model for healthcare can be built by combining multiple data sources and managing its quality to a certain degree. Second, there are identifiable *patterns* for how such a model can be applied in real-world applications.

**Feasibility of a general-purpose statistical model.**   The VHGM is provided as an API and is intended to be a building block for healthcare applications developed and operated by independent vendors. We demonstrate two such applications.

1. **App for encouraging more walking:** A mobile phone company has developed a healthcare app for their phones, which encourages users to walk more for their health. One of the challenges for the app is setting appropriate goals (the number of steps a user should walk daily) because different people have different abilities. The VHGM includes attributes such as daily walking steps and other factors like lower back pain. The app uses this information to suggest, "Individuals with similar profiles but without back pain walk an average of X steps per day," allowing users to decide whether to try walking more.

2. **App for health-related financial assets:** The VHGM contains attributes derived from health insurance records. With these attributes, one can estimate the distribution of annual medical spending, given available attribute values such as age, sex, weight, and lifestyle habits. A financial service startup uses this information to calculate the estimated lifetime medical spending, referred to as "Health Asset." The app enables users to know their Health Asset and experiment with how this number changes based on different habits, such as exercise, drinking, and smoking. To prevent misunderstanding or misuse, the service clearly states that this application is not a medical device and is not intended for the diagnosis, treatment, or prevention of any disease. Users are advised to interpret the results as informative guidance rather than definitive medical or financial advice.

These applications provide evidence of the usefulness of the VHGM as a general-purpose generative model. More applications of the VHGM are described in the model development paper [11].

**Usage patterns of the VHGM.**   As of the time of writing this paper, there are several paying customers who regularly use the VHGM. Additionally, we conducted a couple of business idea contests, asking for new applications based on the VHGM. These experiences revealed recurring usage patterns of the VHGM.

1. **Estimation of a missing value from known values** – This is the basic function of the VHGM. Given observed values $o_1, o_2, ..., o_m$, VHGM returns the estimated distribution $P(y|o_1, o_2, ..., o_m)$ for the target attribute $y$. This pattern is useful when some attribute is difficult to measure directly (e.g., measuring blood sugar usually requires an invasive process – VHGM provides a means to estimate the blood sugar from other observable attributes).

2. **What-if analysis** (Counter-factual scenario generation) – One can provide counter-factual input to the VHGM. For example, "what would my estimated BMI be if I were

not smoking" is a counter-factual query. This should not be interpreted as a causality, but these queries are useful for considering possible scenarios and planning future course of action. The walking encouragement app described above uses this pattern.

3. **Optimization for a desired output** – One can use the VHGM API to iteratively search possible combinations of values that would yield the desired estimated value of the output attribute. For example, "How can I change my diet to reduce the estimated risk of neuropathic pain" would be answered by optimizing the diet attributes to reduce the estimated number of annual doctor visits related to neuropathic pain. Open-source tools for black-box optimization such as Optuna [39] could be used for such computations.

4. **Exploration of possible factors** – One can explore possible attributes that have some relationship with the target attribute. For example, many older adults are concerned about their body odor but are unaware of the factors that may influence it. By changing the value of the body odor attributes and observing how the other over 2000 attributes respond, one may be able to form a hypothesis on the cause of body odor.

For each of the above examples, there are two "modes" of using the VHGM. One is to use it through the API to directly obtain the query results. The other is to generate synthetic data under given conditions and then use the synthetic data for further analysis. In general, this "indirect" mode of use is not recommended because the resulting analysis may contain both the errors incurred by the VHGM training process and the errors in the second, derivational analysis. This mode is, however, useful for:

- **Educational purposes** because the trainees do not need to have a programming environment for API access
- **Analysis in specific groups** for which no data is available.

This is by no means an exhaustive list. We expect that there will be more innovative VHGM use cases in the future.

**Limitations.** Limitations of the VHGM are as follows:

- **Bias in the training dataset** – Due to the cost, logistical constraints, and clinical study purposes, the populations used for building the training dataset were biased. This may make applications that target different populations (e.g., different racial groups) inappropriate. The external dataset validation indicates this limitation. Certain medical conditions specific to populations or environments outside the scope of the training data may not be appropriately addressed.
- **Comprehensiveness of evaluation** – While we devised a multi-faceted evaluation framework (imputation error, distributional concordance, and scenario-based analysis), it does not comprehensively capture all aspects of model quality. Moreover, due to the general-purpose design of the VHGM, it is not feasible to exhaustively benchmark all possible input-output combinations. This limited comprehensiveness should be considered when interpreting the reported results.
- **Deep stratified analysis** – Due to the available sizes of the source datasets, including that of Data Source A, deeply stratified groups do not have enough records, which may result in an unreliable statistical model.
- **Cross-sectionality** – The training dataset is largely cross-sectional (Data Source C-2 was from interventional studies) and therefore, the VHGM is not capable of predicting the

future. It is theoretically possible to build a model with predictive functions if we have high-dimensional time-series data, but collecting such data is excessively expensive.

- **Correlation vs causality** – *What-if analysis* carries the risk of being interpreted as causal. If the VHGM returns "if you were doing daily exercise, your estimated BMI would be lower," it does not mean that exercise will lower the BMI. Clear communication of how to interpret the output of the VHGM is one of the critical risk factors we identified. Transparency (see Governance process) is one of the mitigation efforts.

## Conclusion

We demonstrated the feasibility of constructing a general-purpose generative model for healthcare data. Our analysis confirms that the model captures key statistical properties, including univariate distributions and bivariate relationships among attributes. Additionally, we presented several real-world applications to highlight the model's practical value.

## Supporting information

**S1 Fig. Histogram of correlation coefficients using the Spearman rank correlation.** The number of combinations of selecting pairs of attributes from 1776 attributes is 1,088,550. For each combination, the Pearson rank correlation coefficient was obtained, and the histogram was created.
(TIF)

**S2 Fig. Typical correlated pairs of each data type in Data Source A.** The $\Phi_K$ correlation coefficient [40] was employed to assess the relationships between pairs of variables across various data types, including numerical, categorical, and ordinal. Each 5-letter code corresponds to an attribute definition. (a) Typical correlated pairs between numerical attributes. The numbers in the titles represent the correlation coefficient of the pairs. (b) Typical correlated pairs between a numerical attribute and a categorical attribute. The numbers in the titles represent the correlation coefficient of the pairs. (c) Typical correlated pairs between categorical attributes. The numbers in the titles represent the correlation coefficient of the pairs.
(TIF)

**S1 Text. Analytical method details on skin surface lipid (SSL)-RNA, intestinal microbiota, and saliva microbiota.** Method details are described in S1 Text.
(DOCX)

**S2 Text. Details on the imputation error calculation methods.** Method details are described in S2 Text.
(DOCX)

**S3 Text. Details on the univariate and bivariate analyses.** Method details are described in S3 Text.
(DOCX)

**S4 Text. Details on the external dataset validation.** Method details are described in S4 Text.
(DOCX)

**S1 Table. Number of male participants.** Decade, N (visit 1), Ratio (%), N (visit 2), Conversion (%), and Ref. (%) correspond to the age group of participants, numbers in each age group, percentages of each age group relative to the total participants, ratios of N (visit 2) to N (visit 1), and ratios of the Japanese population from the recent Japanese official statistics that are available at e-Stat (2019), respectively. At visit 2, conversion rates of the men's 60-69

and ≥ 70 age groups were slightly lower compared to those in younger age groups. This reduction appears to be attributable to the exclusion criteria applied in the study or their health conditions (see the medical history records in S3 Table).
(DOCX)

**S2 Table. Number of female participants.** The column name definitions are the same as S1 Table.
(DOCX)

**S3 Table. Number of diseases under treatment.** Numbers of participants undergoing treatment for each disease.
(DOCX)

**S4 Table. Attribute number, missing rate, and outlier rate of each measurement.** Measurement categories were described elsewhere [13].
(DOCX)

## Acknowledgments

The authors thank Koki Tsuda, Satoru Mochizuki, Seoyoun Chung, Takahito Nakamura, Juntaro Yamashita, Tetsuya Uchiyama, Kunihiko Miyoshi, Masahiro Hirasawa, Tsukasa Takemura, Takahiro Takamuku, Tetsuya Yamaguchi, Aiko Suzuki, Shota Hayashi, and Kohei Hayashi for their support for the development of the models and applications. We also thank Kei Sugitani, Adeline Muliandi, Nami Yamanaka, Takahiro Hasumura, Yasutoshi Ando, Takashi Fushimi, Teruhisa Fujimatsu, Tomoki Akatsu, Sawako Kawano, Ren Kimura, Shigeki Tsuchiya, Yuuki Yamamoto, Mai Haneoka, Ken Kushida, Tomoki Hideshima, Eri Shimizu, Jumpei Suzuki, Aya Kirino, Hisashi Tsujimura, Shun Nakamura, Takashi Sakamoto, Yuki Tazoe, Masayuki Yabuki, Shinobu Nagase, Tamaki Hirano, Reiko Fukuda, Yukari Yamashiro, Yoshinao Nagashima, Nobutoshi Ojima, Motoki Sudo, Naoki Oya, Yoshihiko Minegishi, and Koichi Misawa for their analysis to identify the numerous health attributes across diverse categories.

We acknowledge the valuable contributions of Professor Kazuhiro Minami at The Institute of Statistical Mathematics (Tokyo, Japan) through insightful discussions on privacy issues.

We are deeply grateful to MinaCare Co., Ltd. and its founder, Yuji Yamamoto, for providing the commercial healthcare dataset under flexible terms and conditions. Without their belief in the positive impact of widespread data dissemination on healthcare, this project would not have been realized.

## Author contributions

**Conceptualization:** Hiroshi Maruyama.

**Data curation:** Kotatsu Bito, Masanobu Hibi, Shun Katada, Aya Kawakami.

**Formal analysis:** Hiroshi Maruyama, Kotatsu Bito, Yuki Saito.

**Investigation:** Hiroshi Maruyama, Masanobu Hibi, Shun Katada, Kenta Oono, Nontawat Charoenphakdee, Zhengyan Gao, Hideyoshi Igata, Masashi Yoshikawa, Shoichiro Yamaguchi, Yohei Sugawara, Shin-ichi Maeda.

**Methodology:** Masanobu Hibi, Shun Katada, Kenta Oono, Nontawat Charoenphakdee, Zhengyan Gao, Hideyoshi Igata, Masashi Yoshikawa, Shoichiro Yamaguchi, Yohei Sugawara, Shin-ichi Maeda.

**Software:** Kenta Oono, Nontawat Charoenphakdee, Zhengyan Gao, Hideyoshi Igata, Masashi Yoshikawa, Yoshiaki Ota, Hiroki Okui, Kei Akita, Shoichiro Yamaguchi, Yohei Sugawara, Shin-ichi Maeda.

**Supervision:** Hiroshi Maruyama.

**Validation:** Yuki Saito.

**Writing - original draft:** Hiroshi Maruyama, Kotatsu Bito, Yuki Saito, Masanobu Hibi.

**Writing - review & editing:** Hiroshi Maruyama, Kotatsu Bito, Yuki Saito, Masanobu Hibi, Kenta Oono, Nontawat Charoenphakdee, Zhengyan Gao, Hideyoshi Igata.

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
