## [Decision Letter · Decision Letter 0]

20 Jun 2025

PDIG-D-25-00094Creating a General-Purpose Generative Model for Healthcare Data based on Multiple Clinical StudiesPLOS Digital Health Dear Dr. Maruyama, Thank you for submitting your manuscript to PLOS Digital Health. After careful consideration, we feel that it has merit but does not fully meet PLOS Digital Health's publication criteria as it currently stands. Therefore, we invite you to submit a revised version of the manuscript that addresses the points raised during the review process. Please submit your revised manuscript within 60 days Aug 19 2025 11:59PM. If you will need more time than this to complete your revisions, please reply to this message or contact the journal office at digitalhealth@plos.org. Please include the following items when submitting your revised manuscript:* A rebuttal letter that responds to each point raised by the editor and reviewer(s). You should upload this letter as a separate file labeled 'Response to Reviewers'. This file does not need to include responses to any formatting updates and technical items listed in the 'Journal Requirements' section below.* A marked-up copy of your manuscript that highlights changes made to the original version. You should upload this as a separate file labeled 'Revised Manuscript with Track Changes'.* An unmarked version of your revised paper without tracked changes. You should upload this as a separate file labeled 'Manuscript'. If you would like to make changes to your financial disclosure, competing interests statement, or data availability statement, please make these updates within the submission form at the time of resubmission. Guidelines for resubmitting your figure files are available below the reviewer comments at the end of this letter. We look forward to receiving your revised manuscript. Kind regards, Jasmit Shah, PhDGuest EditorPLOS Digital Health Jasmit ShahGuest EditorPLOS Digital Health Leo Anthony CeliEditor-in-ChiefPLOS Digital Healthorcid.org/0000-0001-6712-6626 **Journal Requirements:**

1. We ask that a manuscript source file is provided at Revision. Please upload your manuscript file as a .doc, .docx, .rtf or .tex.

 **Additional Editor Comments (if provided):****Reviewers' Comments:** Reviewer's Responses to Questions

**Comments to the Author**

1. Does this manuscript meet PLOS Digital Health’s publication criteria? Is the manuscript technically sound, and do the data support the conclusions? The manuscript must describe methodologically and ethically rigorous research with conclusions that are appropriately drawn based on the data presented.

Reviewer #1: Yes

Reviewer #2: Yes

2. Has the statistical analysis been performed appropriately and rigorously?

Reviewer #1: No

Reviewer #2: Yes

3. Have the authors made all data underlying the findings in their manuscript fully available (please refer to the Data Availability Statement at the start of the manuscript PDF file)?

Reviewer #1: Yes

Reviewer #2: No

4. Is the manuscript presented in an intelligible fashion and written in standard English?

Reviewer #1: Yes

Reviewer #2: Yes

5. Review Comments to the Author

Reviewer #1: This paper introduces the Virtual Human Generative Model (VHGM), a general-purpose generative model trained on multiple clinical datasets. The model integrates more than 2000 health attributes from various sources to support missing value imputation, synthetic data generation, and exploratory analysis. It is deployed as a commercial API and designed to support personalized and predictive health applications. While the concept is impactful and timely, the scientific rigor and clarity require significant improvement.

Reviewer #2: The authors present an ambitious approach to healthcare data accessibility challenges by producing a general purpose Virtual Human Generative Model. They integrated data from multiple clinical studies including thousands of parameters with intent to implement as an API for personalized and predictive medicine. The methods are robust and the authors did endeavor to address quality assurance and governance related to healthcare AI. Some notable considerations for publication include generalizability, equity, lack of data transparency, and privacy.

The authors acknowledged that the model was trained using data predominately from the Japanese population which significantly limits its external validity warranting a bit more discussion. Some medical conditions are of course affected more than others by this limitation, but, as an example, infectious diseases endemic to areas outside of Japan would be an impossibility to address. The data also seems to be skewed towards chronic conditions as well, so limitations related to acute conditions might need to be commented. What was the rationale for threshold of 0.7 SD for imputation errors? Even this error threshold can have impacts on clinical outcomes and bias towards underrepresented groups, especially when the external validity is already impaired by sample selection. The training algorithm is referenced, but some comment on the core architecture would strengthen the foundation quite a bit. What were some of the details about the process for consent to share data? Not being able to see the substance of the underlying data presents its own limitations.

The inclusion of a governance committee was a welcome addition! Some details about its composition would be helpful, including patient representatives, clinical expertise, ethicists, etc. I do have significant concerns with the "Health Asset" tool presented, though. If this is the intended use case then the ethical nuances should be discussed in more detail. It does appear to be innocuous on the surface with intent to inform users, but, depending on the country adopting its implementation, insurance companies, employers, governments, and others might use this information with intent to discriminate or worsen health disparities for a marginal cost benefit. Other discussion around education and safeguards for the end user would strengthen the manuscript significantly. The methods written do describe important underlying features of the training data such as the univariate and bivariate distributions, but descriptions of other downstream tasks are lacking. If the model is fully intended for prediction tasks then quantifying this with precision, recall, global accuracy, etc would be helpful to read.

Overall the paper is a considerable contribution to the field of digital health. The scale of this approach is a highly valuable contribution to the field. With that being said, more robust solutions require all the more rigor and care when demonstrating consideration of ethics and health equity. Addressing the above points will substantially strengthen the manuscript and ensure that the VHGM is developed and implemented responsibly and equitably.

6. PLOS authors have the option to publish the peer review history of their article (what does this mean?). If published, this will include your full peer review and any attached files.

**Do you want your identity to be public for this peer review?** For information about this choice, including consent withdrawal, please see our Privacy Policy.

Reviewer #1: No

Reviewer #2: **Yes: **Catherine Grace Bielick

---

## [Decision Letter · Decision Letter 1]

6 Oct 2025

Creating a General-Purpose Generative Model for Healthcare Data based on Multiple Clinical Studies

PDIG-D-25-00094R1

Dear Dr. Maruyama,

We're pleased to inform you that your manuscript has been judged scientifically suitable for publication and will be formally accepted for publication once it meets all outstanding technical requirements.

Within one week, you'll receive an e-mail detailing the required amendments. When these have been addressed, you'll receive a formal acceptance letter and your manuscript will be scheduled for publication.

An invoice for payment will follow shortly after the formal acceptance. To ensure an efficient process, please log into Editorial Manager at https://www.editorialmanager.com/pdig/ click the 'Update My Information' link at the top of the page, and double check that your user information is up-to-date. For billing related questions, please contact billing support at https://plos.my.site.com/s/.

Kind regards,

Jasmit Shah, PhD

Guest Editor

PLOS Digital Health

Additional Editor Comments (optional):

Reviewers' comments:

Reviewer's Responses to Questions

**Comments to the Author**

1. If the authors have adequately addressed your comments raised in a previous round of review and you feel that this manuscript is now acceptable for publication, you may indicate that here to bypass the “Comments to the Author” section, enter your conflict of interest statement in the “Confidential to Editor” section, and submit your "Accept" recommendation.

Reviewer #1: (No Response)

Reviewer #3: All comments have been addressed

2. Does this manuscript meet PLOS Digital Health’s publication criteria? Is the manuscript technically sound, and do the data support the conclusions? The manuscript must describe methodologically and ethically rigorous research with conclusions that are appropriately drawn based on the data presented.

Reviewer #1: (No Response)

Reviewer #3: Yes

3. Has the statistical analysis been performed appropriately and rigorously?

Reviewer #1: (No Response)

Reviewer #3: Yes

4. Have the authors made all data underlying the findings in their manuscript fully available (please refer to the Data Availability Statement at the start of the manuscript PDF file)?

Reviewer #1: Yes

Reviewer #3: Yes

5. Is the manuscript presented in an intelligible fashion and written in standard English?

PLOS Digital Health does not copyedit accepted manuscripts, so the language in submitted articles must be clear, correct, and unambiguous. Any typographical or grammatical errors should be corrected at revision, so please note any specific errors here.

Reviewer #1: Yes

Reviewer #3: Yes

6. Review Comments to the Author

Please use the space provided to explain your answers to the questions above. You may also include additional comments for the author, including concerns about dual publication, research ethics, or publication ethics. (Please upload your review as an attachment if it exceeds 20,000 characters)

Reviewer #1: (No Response)

Reviewer #3: I appreciate the authors' efforts in revision, and my concerns have been addressed. This work is I think ready for final acceptance.

7. PLOS authors have the option to publish the peer review history of their article (what does this mean?). If published, this will include your full peer review and any attached files.

**Do you want your identity to be public for this peer review?** For information about this choice, including consent withdrawal, please see our Privacy Policy.

Reviewer #1: None

Reviewer #3: No
